# Hydrocarbon-Stapled Peptide Based-Nanoparticles for siRNA Delivery

**DOI:** 10.3390/nano10122334

**Published:** 2020-11-25

**Authors:** Matthieu Simon, Nabila Laroui, Marianne Heyraud, Guillaume Laconde, Lamiaa M. A. Ali, Kevin Bourbiaux, Gilles Subra, Lubomir L. Vezenkov, Baptiste Legrand, Muriel Amblard, Nadir Bettache

**Affiliations:** 1IBMM, Univ Montpellier, CNRS, ENSCM, 34093 Montpellier, France; matthieusimon.work@gmail.com (M.S.); nabila.laroui@etu.umontpellier.fr (N.L.); marianneheyraud@hotmail.fr (M.H.); guillaumelaconde@gmail.com (G.L.); lamiaa.ali@umontpellier.fr (L.M.A.A.); kevin.bourbiaux@gmail.com (K.B.); gilles.subra@umontpellier.fr (G.S.); lubomir.vezenkov@enscm.fr (L.L.V.); baptiste.legrand@umontpellier.fr (B.L.); 2Department of Biochemistry, Medical Research Institute, University of Alexandria, Alexandria 21561, Egypt

**Keywords:** CPP, Stapled peptides, siRNA, gene silencing

## Abstract

Small interfering RNAs (siRNAs) are promising molecules for developing new therapies based on gene silencing; however, their delivery into cells remains an issue. In this study, we took advantage of stapled peptide technology that has emerged as a valuable strategy to render natural peptides more structured, resistant to protease degradation and more bioavailable, to develop short carriers for siRNA delivery. From the pool of stapled peptides that we have designed and synthesized, we identified non-toxic vectors that were able to efficiently encapsulate siRNA, transport them into the cell and induce gene silencing. Remarkably, the most efficient stapled peptide (JMV6582), is composed of only eight amino-acids and contains only two cationic charges.

## 1. Introduction

Small interfering RNA (siRNA) represents an interesting class of molecules that specifically target and downregulate their mRNA and the corresponding expressed proteins. The ability of siRNAs to “switch off” specific genes makes them attractive to develop new highly specific therapies. Unfortunately, siRNAs are anionic molecules that are unable to cross the cell membrane by themselves because of electrostatic repulsion with the anionic phospholipids of the biological membranes. Furthermore, naked siRNA molecules are rapidly degraded by serum RNases, preventing their direct blood injection as therapeutic treatment [1,2]. Thus, one of the main current challenges to develop siRNA therapies is to improve their delivery into the cytosol while preventing their degradation. Indeed, many different siRNA delivery systems have been investigated such as viral vectors, liposomes, micelles, polymeric nanoparticles, and cell penetrating peptides (CPPs) [3,4,5]. In this study, we focused on the last group of cargo carriers, CPPs, which represent short peptides composed of less than 30 amino acids. CPPs are often amphipathic oligomers with positively charged residues (arginine or lysine), allowing them to enter into the cell and drive the internalization of a wide range of bioactive cargos such as nanoparticles, RNA, proteins or cytotoxic agents. In most cases, the cell delivery of such peptides is improved by their folding, as it was previously observed for some CPPs that naturally adopt a helical fold when they interact with the biological membranes [3,4,5,6,7,8]. More recently, it has been demonstrated that conformational constraints of amphipathic peptides, through multicyclization [9,10] or introduction of α, α-disubstituted residues in the sequence as helical promoters [11,12], can dramatically increase their cellular uptake.

Therefore, in this study we decided to further expand and explore the domain of structured cell penetrating peptide by designing and synthesizing helical peptides for cytosolic delivery. However, the amino acid sequence of natural peptides, even if similar to a natural protein helix, is not sufficient to allow its proper α-helix folding. One strategy to overcome this entropic penalty and induce peptide folding into a biologically active α-helix is the introduction of an all-hydrocarbon staple in the peptide sequence, through ring-closing metathesis (RCM) reaction [13,14,15]. This stapling system, further developed by the Verdine and Walensky groups [16,17,18,19,20,21,22], has been shown to significantly increase α-helical content in short peptide sequences, binding affinity, resistance to proteolysis and importantly to promote efficient cellular uptake [17,20]. The stapled peptide strategy is currently considered as the “gold standard” [23] to stabilize an α-helix and turn a peptide into a biologically active molecule. It allowed the emergence of a new class of biomolecules as suitable pharmacological drug candidates [21,24,25,26,27,28]. A recent publication by the group of Jaehoon Yu provided an elegant proof of concept for the siRNA encapsulation and internalization by Histidine rich cationic stapled peptides [26].

In this study, by taking advantage of the property of hydrocarbon-stapled peptide to internalize into cells, short α-helical peptide sequences were carefully designed and synthesized. After confirmation of their helical folding, we demonstrated their ability to transport efficiently a fluorophore inside the cells without cytotoxicity. Next, we investigated the capacity of these short α-helical peptides to deliver siRNA molecules, as a highly relevant biological cargo. We were able to efficiently encapsulate siRNA and induce gene silencing in the nanomolar range with a short helical stapled peptide, possessing only eight amino-acids and two cationic charges.

## 2. Materials and Methods

### 2.1. Materials and Reagents

All reagents and solvents were obtained from commercial sources and were used without further purification. Analytical HPLC analyses were run on an Agilent Technology 1220 Infinity LC (Agilent, Santa Clara, CA, USA) equipped with a Chromolith Speed Rod RP-C18 185 Pm column (50 × 4.6 mm, 5 μm) with a gradient from 100% (H_2_O/TFA 0.1%) to 100% (CH_3_CN/TFA 0.1%) in 5 min; flowrate 4 mL/min; UV detection at 214 nm (conditions B). LC/MS analyses were recorded on a Quattro micro ESI triple quadrupole mass spectrometer (Micromass, Manchester, UK) equipped with a Chromolith Speed Rod RP-C18 185 Pm column (50 × 4.6 mm, 5 μm) and an Alliance HPLC System (Waters, Milford, MA, USA); gradient from 100% (H_2_O/HCO_2_H 0.1%) to 100% (CH_3_CN/HCO_2_H 0.1%) in 3 min; flowrate 3 mL/min; UV detection at 214 nm. High-Resolution Mass Spectrometric analyses were performed with a time-of-flight (TOF) mass spectrometer (Waters, Milford, MA, USA) fitted with an electrospray ionization source (ESI) in positive ion mode.

The different siRNA sequences are for anti-firefly luciferase (siLuc): 5′-CUUACGCUGAGUACUUCGAdTdT-3′ (sense strand), and 5′-UCGAAGUACUCAGCGUAAGdTdT-3′ (anti-sense strand) and for siRNA without biological activity, used as control (siCtrl): 5′-CGUACGCGGAAUACUUCGAdTdT-3′ (sense strand) and 5′-UCGAAGUAUUCCGCGUACG dTdT-3′ (anti-sense strand), and the cy5-labelled control siRNA (siCtrl-cy5) were purchased from Eurogentec (Serring, Belgium).

Lipofectamine RNAiMAX reagent was purchased from Invitrogen (Cergy Pontoise, France). Luciferin was purchased from Promega (Charbonnières-les-bains, France). Cell viability reagent 3-(4,5-dimethylthiazol-2-yl)-2,5-diphenyltetrazolium bromide (MTT) was purchased from Sigma-Aldrich (Saint-Quentin-Fallavier, France).

### 2.2. Synthesis and Purification of Peptides and Stapled Peptides

Solid phase peptide synthesis was performed on an Amphispheres^®^ (Agilent, Santa Clara, CA, USA) Rink Amide resin loaded at 0.38 mmol/g using Fmoc/t-Bu chemistry. First, resin was soaked in Dichloromethane (DCM) (Sasu Carlo Erba reagents, Val de Reuil, France) for 10 min and filtered. For each coupling reaction, 5 equivalents (eq.) of Fmoc-Amino Acid, 5 eq. of Hexafluorophosphate Azabenzotriazole Tetramethyl Uronium (HATU) and 10 eq. of *N*,*N*-Diisopropylethylamine (DIEA) were added to a fritted reaction vessel and stirred in *N*,*N*-dimethylformamide (DMF) (2 × 5 min at r.t.). For the coupling of olefinic amino acid, 3 eq. of Fmoc-(S)-pentenylalanine, 3 eq. of HATU and 5 eq. of DIEA were added to the reactor and stirred in DMF (2 × 30 min). Deprotection of the Fmoc group at the N-terminus was performed using a 20% piperidine/DMF (Sasu Carlo Erba reagents, Val de Reuil, France). solution (2 × 5 min at r.t.). After each coupling and deprotection steps, resin was washed three times with DMF. After removal of the Fmoc group of the 6-(Fmoc-amino)hexanoic acid linker (Ahx), peptides were labelled by using 2 eq. of fluorescein isothiocyanate and 4 eq. of DIEA in DMF and stirred for 3 h.

Ring closing metathesis was directly performed on the solid support for peptides containing (S)-pentenylalanine residues in their sequence by using 0.4 eq. of Grubbs catalyst (first generation) in 1,2 dichloroethane, under inert atmosphere (stirred 2 × 2 h at r.t.), then resin was washed with 3 × DCM, 3 × DMF, 3 × DMF.

Peptides were then cleaved from the resin with a TFA/TIS/H_2_O 95/2.5/2.5 vvv solution (2 × 90 min at r.t.). Resins were washed (1 × DCM, 1 × TFA, 1 × DCM) and filtrates were evaporated under reduced pressure. Compounds were precipitated by addition of 50 mL of diethyl ether (Sasu Carlo Erba reagents, Val de Reuil, France). and centrifuged (3000 rpm, 20 min). Crude peptides were then solubilized in acetonitrile/water 1/1 *vv* solution and purified by preparative RP-HPLC (Waters, Milford, MA, USA) on a Waters system controller equipped with a C18 Waters Delta-Pack column (100 × 40 mm, 100 Å); flowrate 50 mL/min; UV detection at 214 nm using a Waters 486 Tunable Absorbance Detector and a linear gradient of A = H_2_O (0.1% TFA) and B = CH_3_CN (0.1% TFA). Peptides were recovered as TFA salts. Final compound purity was assessed by LC-MS analysis.

### 2.3. Circular Dichroism (CD) and Determination of Helical Content

Circular dichroism experiments were carried out using a Jasco J815 spectropolarimeter. Spectra (Jasco, Easton, MD, USA) were typically recorded with 100 μM of peptides dissolved in a water/TFE mixture (95/5), pH 7, using a 1 mm pathlength CD cuvette at 20 °C, over a wavelength range of 190–260 nm. The peptide concentrations were normalized measuring the OD at 490 nm considering a molar coefficient extinction (ε_490_) for the FITC of 76,900 cm^−1^ M^−1^. Continuous scanning mode was used, with a response of 1.0 s with 0.1 nm steps and a bandwidth of 2 nm. The signal to noise ratio was improved by acquiring each spectrum over an average of two scans. Baseline was corrected by subtracting the background from the sample spectrum. Alpha helical content was determined using the following equation: % Helix = ([θ])obs × 100)/(−39,500 × (1 − 2.57/n), where ([θ])obs is the mean residue ellipticity at 222 nm and n the number of peptide bonds [28].

### 2.4. Gel Retardation Assay

In RNase free water, a fixed concentration, 100 picomoles, of siCtrl was mixed with the appropriate amounts of stapled peptide in order to reach an amine-phosphate (N/P) ratio of 2, 5, 10, and 20. Blue 6X loading dye (Fisher Scientific) was added to the mixture. Electrophoresis was carried out on a 2% *w*/*v* agarose gel mixed with GelRed^TM^ nucleic acid gel stain (Interchim, Montluçon, France) in 1X TBE buffer (90 mM Tris-borate/2 mM EDTA, pH 8.2). The gel was run in 0.5X TBE at 50 V for 1 h. A 100 bp DNA ladder from Sigma-Aldrich (Saint-Quentin-Fallavier, S4 France) was used as a reference for the gel. The GelRed-stained siRNA was visualized using a TFX-20 M model-UV transilluminator (Vilber Lourmat, Marne-la-Vallée, France) and gel photographs were obtained with a smartphone camera

### 2.5. Dynamic Light Scattering (DLS) and Zeta Potential Measurements

The stapled peptides/siRNA complexes were prepared using siCtrl as previously described to achieve the desired N/P ratio, and then were diluted in glucose 5%. Measurements were performed immediately after the complexes formation using Zetasizer Nano-ZS instrument (Malvern Instruments Ltd., Worcestershire, UK) with transparent ZEN0040 disposable micro-cuvette at 25 °C. For Zeta potential analyses, complexes were prepared as previously described and then diluted in glucose 5% and NaCl (5 mM, pH = 7.0) to obtain a final volume of 1 mL. Measurements were performed using Zetasizer Nano-ZS instrument (Malvern, UK) and DTS 1070 zeta potential cells (Malvern Instruments Ltd., Worcestershire, UK) at 25 °C. Three measurements were made with 12 runs for each.

### 2.6. Cell Culture

Human breast adenocarcinoma (MDA-MB-231, ATCC^®^ CRM-HTB-26™) cell line and HeLa human cervix cancer cells (ATCC^®^ CCL-2™) were purchased from ATCC (Manassas, VA, USA). MDA-MB-231-Luc-RFP stable cell line was obtained from AMSBIO (SC041, Abingdon, UK). Cells were grown in Dulbecco’s modified Eagle’s medium (DMEM) supplemented with 10% fetal bovine serum (Invitrogen, Cergy Pontoise, France) and gentamycin (0.05 mg mL^−1^) (Invitrogen, Cergy Pontoise, France) and incubated at 37 °C in a humidified atmosphere with 5% CO_2_.

### 2.7. Cell Viability (MTT) Assay

MTT (4,5-dimethylthiazol-2-yl)-2,5-diphenyltetrazolium bromide) (ThermoFisher Scientific, Courtaboeuf, France) assay was performed to evaluate the cell viability [29]. Briefly, 2000 cells were seeded into a 96 multi-well plate in 200 μL complete culture medium. Twenty four hours after seeding, increasing concentrations (from 0 to 20 µM) of stapled peptides were added to cells with or without an excess of 10 equivalent of Dithiothreitol (DTT) (Sigma Aldrich Chimie Sarl, Saint Quentin Fallavier, France) (from 0 to 200 µM) for 72 h. Cells treated with the vehicle were considered as a control. After this incubation, cells were treated for 4 h with 0.5 mg mL^−1^ of MTT in media. The MTT/media solution was then removed and the precipitated crystals were dissolved in EtOH/DMSO (1:1). The solution absorbance was read at 540 nm. The percentage of viable cells were calculated according to the following equation (Ab test/Ab control × 100).

### 2.8. Cellular Uptake of FITC-Stapled Peptides

MDA-MB-231 cells were seeded in black 96-well plate in 200 μL of their corresponding medium. After 24 h of cell growth, cells were treated with 5 µM concentration of FITC-stapled peptides for 4 h. Total fluorescence (TF) was measured using CLARIOstar^®^ High Performance Monochromator Multimode Microplate Reader (BMG Labtech, Ortenberg, Germany) at excitation/emission of 483-14/530-30 nm. Cells were washed three times with PBS and remaining fluorescence (RF) was measured using the same conditions. The percentage of uptake values were presented as relative mean fluorescence and normalized with respect to Penetratin used as a standard control and set at 100%.

### 2.9. Imaging

One day prior to the experiment, Hela cells were seeded into 12 well plate containing an 18 mm diameter glass coverslip (ThermoFisher Scientific, Courtaboeuf, France) at a density of 2000 cells cm^−2^ in their corresponding medium. Twenty-four hours after cell growth, adherent cells were washed once with DPBS and incubated in 1 mL culture medium containing complex stapled peptides/siCtrl-cy5 (100 nM concentration of siRNA) at N/P = 5 for JMV6582 and N/P = 2 for JMV6580 and JMV6583, stapled peptides alone, Lipofectamine/siCtrl-cy5 (50 nM) and siCtrl-cy5 alone (100 nM) for 4 h. After incubation, adherent cells were washed and fixed with 3.7% paraformaldehyde (PFA). Prior to the microscopic observation, cells were washed gently 3 times with DPBS, then incubated with DPBS containing CellMask^TM^ orange plasma membrane stain (Invitrogen, Cergy Pontoise, France) and Hoechst 33,342 (Invitrogen, Cergy Pontoise, France) at a final concentration of 5 μg mL^−1^ and 10 μg mL^−1^, respectively for 15 min. Cells were washed twice then were visualized with a Zeiss LSM780 confocal microscope (Carl Zeiss Microscope, France) at 649 nm for siCtrl-cy5, 488 nm for stapled peptide, 561 nm for cell membranes and 360 nm for Hoechst 33342 at high magnification (40×/1.4 OIL Plan-Apo).

### 2.10. Cell Luciferase Assay

MDA-MB-231-Luc-RFP cells were seeded at a density of 2000 cells per well in 96-well white opaque tissue culture plates in 200 μL of phenol red free DMEM medium. Twenty four hours after, cells were washed with PBS and incubated with stapled peptide/siLuc complex formulations at the appropriate N/P ratio with the siLuc concentrations of 50, 100, and 200 nM at 37 °C for 4 h. Thereafter, 50 μL of 40% serum containing medium was added to reach the concentration of 10% serum and a final volume of 200 μL. Lipofectamine RNAiMAX Reagent (Invitrogen, Cergy Pontoise, France) was used as the positive control transfection reagent following the manufacturer’s protocol. Three days after transfection, expression of luciferase was assessed by addition of luciferin (10^−3^ M, final concentration) into culture medium. After 10 min, living cell luminescence was measured using a plate reader CLARIOstar^®^ (BMG Labtech, Ortenberg, Germany) at 562 nm and were averaged from triplicates. The percentage of luminescence of treated cells was calculated by using the control cell as 100%. Each assay was repeated three times. Luciferase activity was normalized in accordance to the total number of living cells in each sample as determined by the MTT assay as previously described.

### 2.11. Statistical Analysis

Each assay was repeated three times. Data are presented as the mean ± standard deviation (SD). Statistical analysis was performed using GraphPad Prism 5 (GraphPad Software, Inc., San Diego, CA, USA). The comparison between groups was analyzed with Student’s *t*-test. Differences were considered statistically significant when p values were less than 0.05 (*p* < 0.05). The level of significance was defined at ns (not significant), * *p* < 0.05, ** *p* < 0.01, and *** *p* < 0.001.

## 3. Results and Discussion

### 3.1. Design, Synthesis, and Characterization of the CPPs

In this study, we focused on the design and selection of short amphipathic peptide sequences (less than 10 amino acids) containing an all-hydrocarbon staple for their ability to efficiently enter into cells and to encapsulate siRNA and translocate them to the cells. To achieve this goal, we have built a series of stapled peptides that incorporated cationic and hydrophobic amino acids, in particular Arg and Ile residues, respectively and that adopted a stable helix structure (Figure 1a). Noteworthy, Arg was preferred to Lys residues due to the stronger interaction of the guanidinium group of the Arg residues with the phospholipid head groups of the biological membranes compared to the amino group of the Lys residues, thus resulting in a better internalization [30]. For the evaluation of the cellular uptake of all peptides, an amino hexanoic acid spacer followed by a fluorescein at their N-terminal extremity was introduced and the well-known CPP Penetratin [31] was also synthesized to be used as a positive control (Figure 1a).

A first stapled peptide, JMV6337 was synthesized. Four arginine residues were introduced in this peptide sequence since it was previously described that best internalization was achieved using stapled peptides with a +3 to +5 formal charge at physiological pH [9]. The hydrophobic part of the peptide was made of a sequence of three isoleucine residues. This sequence was chosen according to the work performed by Matsumoto et al. [32] who showed that the incorporation of a hydrophobic Phe-Phe-Ile or Ile-Ile-Ile sequence increases the potency of their vectors. Even if the sequence Phe-Phe-Ile was shown to be the most effective hydrophobic sequence for internalization, we focused on the Ile-Ile-Ile sequence, in order to prevent steric hindrance between the staple and the phenyl group of the phenylalanine residues.

Finally, in order to induce a stable alpha-helical secondary structure, a property that is given to favor cellular uptake, we introduced a stapling system into the peptide sequence (Figure 1b).

Given the dependence of cell-penetration with helicity but also hydrophobicity, we decided to introduce an all-hydrocarbon stapling according to the Verdine technology [19]. For that purpose, di-substituted amino acid residues were introduced before and after the Isoleucine sequence at positions i, i + 4. Several analogues of the peptide JMV6337 were then synthetized (Figure 1a). First, in order to conjugate the stapled peptides to different cargo molecules through thiol-maleimide chemistry, a Cysteine at the C-terminal extremity of JMV6337 was introduced to lead to peptide JMV6580. Then, its corresponding linear analogue JMV6579, with two Ala residues instead of the di-substituted amino acids, was synthesized in order to assess the importance of the staple for the efficiency of this family of molecules in the internalization process and the cellular uptake of siRNA. We also reduced the number of positive-charged residues by synthesizing an analogue that contains only two Arginine residues (JMV6582). Finally, in order to evaluate the impact of reversing the positions of the hydrophobic and hydrophilic parts inside the sequence, we built the reversed peptide JMV6583 by Arginine and Leucine residues. First, Figure 1c showed that the ruthenium-catalyzed, ring-closing olefin metathesis reaction between the two α-methyl, α-alkenyl glycine residues allowed the induction of helical conformation over the peptide sequences. Indeed, all stapled peptides displayed typical α-helix CD profiles with two negative bands at 208 nm and 222 nm and a strong positive band centered at 195 nm while the linear peptide (JMV6579) remained partially unfolded with two minima at 199 and 226 nm. Despite the short peptide sequences, the incorporation of the hydrocarbon-stapling conferred a significant enhancement of the α-helical content up to 20%.

### 3.2. Cytotoxicity and Cellular Uptake Studies

In a first instance, cell viability in the presence of stapled peptides was evaluated on human breast adenocarcinoma MDA-MB-231 cells for 72 h at different concentrations. Results showed that JMV6579, JMV6580, JMV6582, and JMV6583 had low cytotoxic effect up to 20 µM concentration, with cell viability values of 83 ± 5%, 75 ± 6%, 88 ± 6%, and 77 ± 2%, respectively (Figure 2a). In contrast, JMV6337 exhibited higher cytotoxicity at 5 µM with a cell death percentage value of 70 ± 5%. At higher concentration, we observed a drastic increase of cytotoxicity that reached 93 ± 2% of cell death at 20 µM.

The difference between the non-cytotoxic stapled peptides (JMV6580, JMV6582, and JMV6583) and the cytotoxic one (JMV6337) is the absence of a Cys residue at the C-terminal extremity of the peptide JMV6337. In order to evaluate whether the absence of cytotoxicity was due to the presence of the cysteine residue, responsible for disulfide bond formation, we assessed the cytotoxicity of the stapled peptides in reducing conditions. As shown in Appendix A, the stapled peptides containing cysteine were still non cytotoxic even in the presence of a reducing agent (dithiothreitol). As expected JMV6337 remained cytotoxic and its Cys-containing analogue JMV6580 was moderately cytotoxic. This result showed that the absence of cytotoxicity in cysteine-containing stapled peptides cannot be explained by the dimerization of the stapled peptides via disulfide bond formation as shown by Amand et al. [33] but may be related to a difference in their physicochemical properties. According to these results, the stapled peptide JMV6337 was not selected as potential vectors for binding and shuttling siRNA into cells. Nevertheless, this peptide was tested for siRNA complexation and gene silencing activity. Even complexed to siRNA, this peptide is toxic and has no appreciated luciferase inhibitory effect as shown in Appendix A.

The ability of the stapled peptides to penetrate cells was assessed using fluorescence emission measurements. MDA-MB-231 cells were incubated with FITC-stapled peptides at 5 µM concentration for 4 h. Results showed that all stapled peptides tested were more efficiently internalized than the positive control penetratin (Figure 2b). The peptide JMV6583 showed the highest internalization efficiency with 456 ± 1% relative mean FITC fluorescence compared to 100 ± 5% of fluorescence obtained by penetratin. The stapled peptides JMV6582 and JMV6580 also showed high internalization with the values of 391 ± 8% and 313 ± 2% respectively compared to penetratin internalization. The data showed that all the studied stapled peptides exhibited a good internalization efficiency that was more than 3-fold higher than that of penetratin. The linear peptide JMV6579 was almost not able to enter cells. This result supported that the all-hydrocarbon stapling is a key element for high cellular uptake of this series of molecules.

We then further investigated the ability of the more potent and less toxic stapled peptides, i.e., JMV6580, JMV6582, and JMV6583 to assemble with siRNA and mediate efficient siRNA delivery into cells.

### 3.3. siRNA Delivery Studies

To validate the hypothesis that the positively charged stapled peptides (from 2 to 4 charges) could potentially complex the negatively charged siRNA, we first studied the complexation of the siRNA with the stapled peptide by agarose shift assay (Figure 3a). This assay was used to follow the complexation state in a charge ratio-dependent manner because siRNA migration into the agarose gel will be prevented by cationic stapled interactions. In this experiment, we used siCtrl and tested different N/P ratios (2, 5, 10, and 20). The results showed that there was no complete shift of the siRNA at N/P ratio of 2 for JMV6582 suggesting an only partial complexation. However, a strong retardation effect was observed with a complete disappearance of the band corresponding to the native siRNA at N/P = 5. In contrast, the compounds JMV6580 and JMV6583 were able to complex siCtrl at N/P = 2. In conclusion, agarose gel assay clearly showed that the three tested stapled peptides have the ability to efficiently complex siRNA.

The particle size and the charge of the complex are important factors for cellular uptake and nucleic acid delivery. The condensation of siCtrl into nanoparticles, promoted by stapled peptides, was studied by dynamic light scattering experiment at N/P = 5 for JMV6582 and N/P = 2 for JMV6580 and JMV6583. Size and homogeneity were determined for each complex as shown in Figure 3b. DLS measurements revealed that JMV6582 leads to small nanoparticles with a hydrodynamic diameter of 152 ± 59 nm and polydispersity index (PDI) of 0.68 ± 0.27. However, we observed the formation of much larger particles with the two other compounds JMV6580 and JMV6583 (average diameters were 1720 ± 300 nm and 963 ± 56 nm respectively, with PDI of 0.31 ± 0.13 and 0.51 ± 0.11, respectively). Charge surface of siRNA/stapled peptide complex was also evaluated by zeta potential measurements revealing close values of +119 ± 9, +149 ± 5, and +132 ± 3 mV for JMV6580, JMV6582, and JMV6583 complexes, respectively. The formed complexes are positively charged, which is an important factor for cell entry.

### 3.4. Cellular Uptake

Cellular uptake of the stapled peptide/siRNA complexes was evaluated in Hela cells by fluorescence confocal microscopy as shown in Figure 4. The results were compared with those of the siRNA alone (Free siRNA) and the Lipo/siRNA complex. After 4 h of incubation, while the siRNA alone was not able to enter cells, the Lipo/siRNA complex was able to deliver siRNA into cells. Interestingly, the three selected stapled peptides in complex with siCtrl-cy5 were also able to be internalized into the cells (FITC, green staining) and deliver the siCtrl-cy5 into the cytoplasm (siCtrl-cy5, magenta staining). The cellular uptake of the complexes was quantified by flow cytometry and the results showed similar extent of internalization for all the stapled peptide/siRNA complexes (96.65%, 92.24%, and 83.75% for JMV6580/siRNA, JMV6582/siRNA and JMV6583, respectively; Appendix A. The extent of cellular uptake of the stapled peptide/siRNA complexes was slightly different from the cellular uptake of the stapled peptides alone (Figure 2b). The cytosolic localization of the stapled peptides/siRNA complexes is especially interesting for therapeutic applications, due to the large number of potential targets localized in this compartment. Such small peptides constituted attractive molecules for further investigation in the delivery of bioactive siRNA. As a proof of concept, we investigated their efficiency to inhibit the luciferase activity in MDA-MB-Luc-RFP cell line.

### 3.5. Inhibition of Luciferase Activity in MDA-MB-231-RFP-Luc Cells

As described above and to evaluate the therapeutic potential of these nanovectors for siRNA delivery, we set up experiments using siRNA targeting the expression of luciferase gene in MDA-MB-231-Luc-RFP cells. The results showed that the three stapled peptides/siLuc complexes inhibited luciferase activity in a dose-dependent manner in MDA-MB-231-Luc-RFP cells. (Figure 5). Whereas, both JMV6580/siLuc and JMV6583/siLuc complexes inhibited moderately luciferase activity by 34 ± 9% of and 38 ± 12%, respectively, it is noteworthy that the shortest stapled peptide (JMV6582) with only eight amino-acids and two cationic charges exhibited the highest luciferase inhibition. To test the specificity of the stapled peptides/siRNA complexes to inhibit the luciferase activity, we evaluated the luciferase activity by using a control siRNA (siCtrl). As expected, there is no inhibition of luciferase activity with siCtrl (Appendix A. Importantly, the data showed that all the stapled peptide/siRNA complexes were non-toxic compared to the lipofectamine/siRNA transfection. The efficiency of gene silencing of JMV6582 compared to JMV6580 and JMV6583 may be explained by the difference of the stapled peptide/siRNA particle size. Indeed, the smaller size of JMV6582/siRNA complex (152.2 nm) compared to larger sizes of JMV6580/siRNA and JMV6583/siRNA complexes (1720 nm and 963 nm respectively) is more favorable for transfection efficiency [34].

In consequence, the JMV6582/siRNA formulation has the potential to be further developed as a non-invasive siRNA delivery system since the designed stapled peptides are shown to be non-immunogenic with viable pharmacokinetics and high levels of in vivo stability [35].

## 4. Conclusions

In this study, we have designed and synthesized stapled peptides for the delivery of siRNA into cells. These peptides are no longer than 10 amino acids, they contain only 2 to 4 positively charged amino acids and, except for JMV6337, they contain a C-terminal Cys residue. Surprisingly, this latter peptide exhibited a high toxicity. Our study revealed that all the Cys-containing stapled peptides (JMV6580, JMV6582, and JMV6583) were efficiently internalized into cells. As expected, the data showed that the linear peptide JMV6579 failed to enter cells, supporting the importance of the secondary helical structure for efficient cellular uptake of this family of compounds. Then, we demonstrated that these three stapled peptides were capable of complexing siRNA and forming positively charged nanoparticles of different sizes, leading to the translocation of luciferase siRNA. These conditions favor membrane interactions leading to their internalization inside cells. Indeed, the cell-internalized complexes are efficient to target the selected gene and to inhibit its expression. Remarkably, the most efficient stapled peptide JMV6582 is composed of only eight amino-acids and two cationic charges that constitutes to our knowledge, the smaller peptide that was capable to form non-covalent complexes with siRNA and inhibit the luciferase gene expression with a good efficiency.

Finally, although we used siRNA targeting luciferase to explore the efficacy of this novel delivery system, our results open the way for the use of stapled peptide vectors as relevant siRNA targeting genes involved in several diseases including cancer.

## Figures and Tables

**Figure 1 nanomaterials-10-02334-f001:**
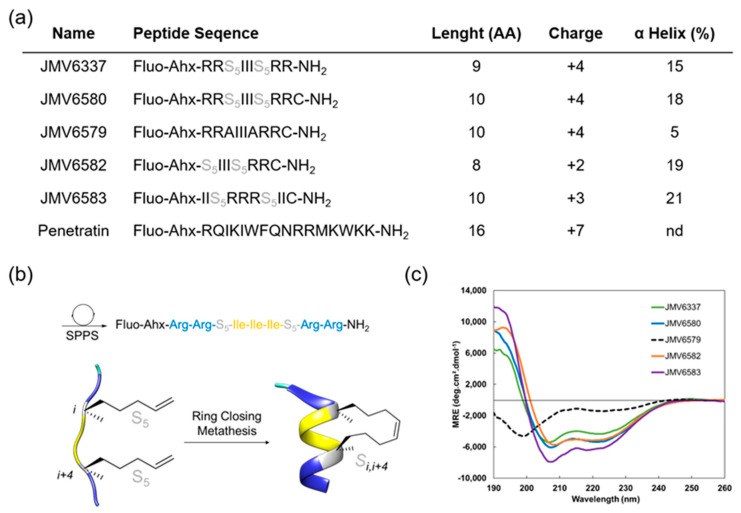
(**a**) Summary of fluorescent peptides used in this study. (**b**) Schematic representation of fluorescent stapled peptide synthesis by ring-closing metathesis (JMV6337), S5 corresponds to the (S)-pentenylalanine residue, (**c**) Overlay of far-UV circular dichroism (CD) spectra of stapled peptides. Experiment were performed at 20 °C in a water/TFE mixture (95/5).

**Figure 2 nanomaterials-10-02334-f002:**
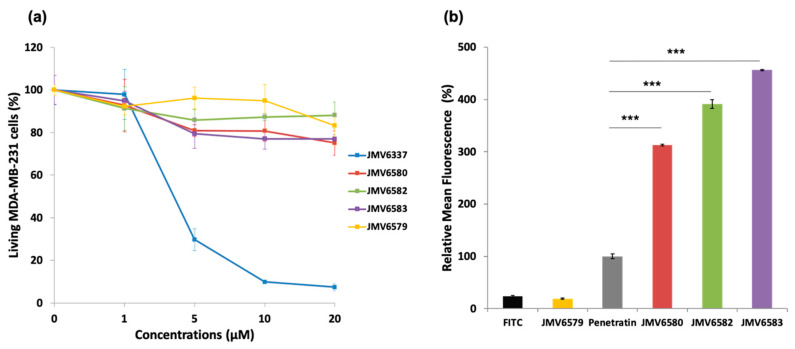
Cytotoxicity and cellular uptake studies of FITC-stapled peptides (**a**) Cytotoxicity study in human breast adenocarcinoma MDA-MB-231 cells incubated with increasing concentrations (from 0 to 20 µM) of the stapled peptides for 72 h. (**b**) Uptake of FITC-stapled peptides at 5 µM final concentration in MDA-MB-231 cells after 4 h of incubation represented as relative mean fluorescence. The values are normalized with respect to penetratin. Results are presented as means ± standard deviations of three independent experiments performed in triplicate.

**Figure 3 nanomaterials-10-02334-f003:**
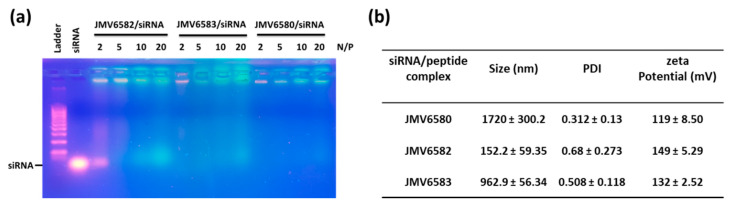
Studies of stapled peptides complexation with siRNA (**a**) Gel electrophoresis analysis for different stapled peptides at N/P = 2, 5, 10, and 20. (**b**) Physicochemical characterization (Particle size, polydispersity index (PDI) and ζ-potential measurements) of stapled peptide/siRNA complex at N/P = 2 for JMV6580 and JMV6583, and N/P = 5 for JMV6582. The values are represented as mean ± SD (n = 3).

**Figure 4 nanomaterials-10-02334-f004:**
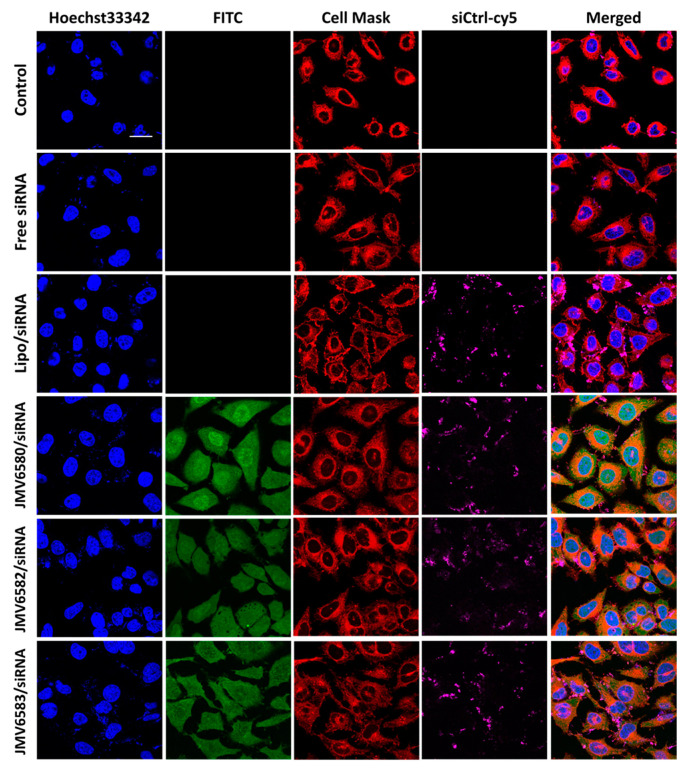
Cellular uptake of stapled peptides/siRNA complexes in HeLa cells visualized by fluorescence confocal microscopy. From the top to the bottom respectively: Untreated cells, cells incubated with 100 nM of siCtrl-cy5, cells incubated with the lipofectamine/siCtrl-cy5 complex at 50 nM, and finally cells incubated with the complex formed between stapled peptides (JMV6580, JMV6582 and JMV6583 respectively) and siCtrl-cy5 (100 nM) at N/P = 2 for JMV6580 and JMV6583, and N/P = 5 for JMV6582. The blue fluorescence indicates the nuclei, stapled peptides appear in green color, cell mask appears in red color, siCtrl-cy5 appears in magenta color. Scale bar = 20 μm.

**Figure 5 nanomaterials-10-02334-f005:**
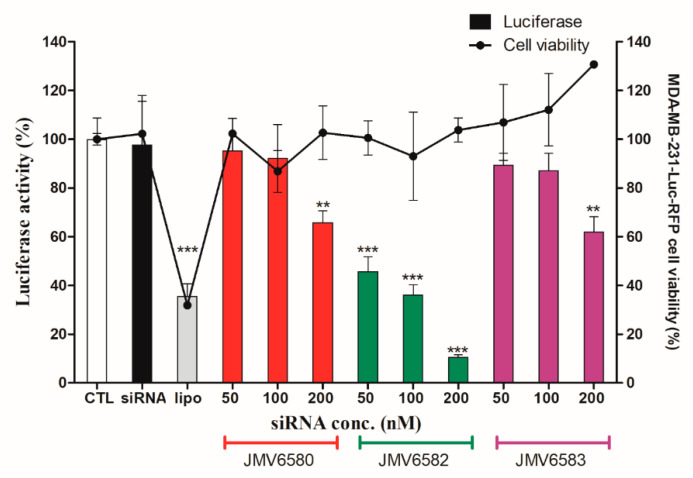
Luciferase activity assay showing the transfection of a 21-mer small interfering RNAs (siRNAs) targeting the expression of luciferase inside MDA-MB-231-Luc-RFP cells. The experiments were carried out with increasing amounts of siLuc (from 50 to 200 nM) complexed with the stapled peptides. The corresponding concentrations of the peptides are from 1.05 µM to 4.2 µM for JMV6580, from 1.4 µM to 5.6 µM for JMV6583 and from 5.25 µM to 21 µM for JMV6582. For lipofectamine transfection, the siRNA concentration used is 50 nM. In parallel, cell viability was measured for each condition. Results are expressed as mean ± standard deviation (n = 3). Statistically different, the level of significance was defined ** *p* < 0.01, and *** *p* < 0.001.

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
