# Peer review of "Hydrocarbon-Stapled Peptide Based-Nanoparticles for siRNA Delivery"

_nanomaterials, 2020, doi:10.3390/nano10122334_

Round 1

Reviewer 1 Report

This paper describes a systematically designed effort to develop short stapled peptide sequences that can be used for safe and efficient delivery of siRNA therapeutics. The introduction is relatively well constructed to deliver the necessary background, rationale, and motivation. And the experimental designs are good to test some suggested hypotheses. However, some part of the results and conclusions are not fully supported by the data, because of the lack of experiments and/or presentation of such. Overall, the paper has novel findings that can be an interest to the field. However, it requires some major revisions before publication in Nanomaterials:

Major revisions:

  1. The overall results indicate that the presence of C-terminal Cysteine residue potentially play a significant role in the cytotoxicity, internalization capability, and the complexation of peptide and siRNA. However, this effect has been only partially tested by the initial cytotoxicity test (Figure 2), and the peptide JMV6337 (the only peptide without the Cys) was ruled out from all the other subsequent experiments. This aspect must be strengthened by the following efforts.
  • In order to confirm their interpretation of cytotoxicity data properly (Lines 267-271), they have to try the same experiments (MTT assay) under a reducing condition (e.g. in the presence of beta-mercaptoethanol or DTT) to see if indeed the dimerization by disulfide bonds plays in the reduction of cytotoxicity.
  • And in order to test and examine if all the subsequent data coming from Peptide 6580, 6582 and 6583 are obtained by the dimeric peptides with disulfide bonds, the same analogs of peptide 6580, 6582, and 6583 without the Cys residue need to be synthesized and tested together alongside.
  1. Figure 1c shows the CD spectrum of peptide 6337 to confirm that the stapled structure promotes the alpha-helix formation. As this becomes the main premise of the paper for interpretation of the rest of the experiments and results, the same CD measurements should be performed and presented for all the other peptides as well (particularly important to compare with the peptide 6579 that has no stapled structure). If the data is too big to be shown in the main figure, the authors might be able to put it in the Supplementary Information.
  2. In Figure 4, the cells treated by peptide 6582 and Lipofectamine (those groups that show the highest siRNA efficacies) look much unhealthier than the other groups. Authors need to explain this, probably in relation to the data (cell viability) presented in Figure 5.
  3. In Figure 5, what was the dosage of siRNA in the lipofectamine group (positive control). It should be clearly mentioned and discussed in the text and the figure caption, because the results from the peptide show dose-dependent behaviors.

Minor:

  1. Line 38: The first appearance of abbreviation needs the full name description. CPP -> Cell Penetrating Peptide.
  2. Table 3b: comma (,) needs to be replaced by (.) in numbers

Author Response

We thank the reviewer for his valuable comments and suggestions which are helpful to improve our manuscript.

Please find below our answers to your comments and suggestions:

Major revisions: 

  1. The overall results indicate that the presence of C-terminal Cysteine residue potentially play a significant role in the cytotoxicity, internalization capability, and the complexation of peptide and siRNA. However, this effect has been only partially tested by the initial cytotoxicity test (Figure 2), and the peptide JMV6337 (the only peptide without the Cys) was ruled out from all the other subsequent experiments. This aspect must be strengthened by the following efforts.
  • In order to confirm their interpretation of cytotoxicity data properly (Lines 267-271), they have to try the same experiments (MTT assay) under a reducing condition (e.g. in the presence of beta-mercaptoethanol or DTT) to see if indeed the dimerization by disulfide bonds plays in the reduction of cytotoxicity.

We agree with the reviewer. To elucidate whether the absence of cytotoxicity is related to the disulfide bonds, we have done the cytotoxic assay in the presence of a reducing agent (DTT). We completed the profile of cell viability, presented in Figure 2a, in reducing conditions, and as shown in Figure S1, no cytotoxicity was observed in reducing conditions (see Figure S1 in Supplementary Material). We concluded from this experiment, that the absence of cytotoxicity was not related to the dimerization but probably to a difference in physicochemical properties. This result is now included in the text (Lines 270-278, page 7).

  • And in order to test and examine if all the subsequent data coming from Peptide 6580, 6582 and 6583 are obtained by the dimeric peptides with disulfide bonds, the same analogs of peptide 6580, 6582, and 6583 without the Cys residue need to be synthesized and tested together alongside.

In absolute way, we agree with the comment of the reviewer. But the synthesis of the stapled peptides takes time. Indeed, a response to this comment is partially provided with a couple of stapled peptides explored in this manuscript. The stapled peptide JMV6580 has the same sequence as JMV6337 with a cysteine at its C-terminal. Furthermore, the cytotoxicity experiment in the presence of reducing agent showed that the absence of cytotoxicity was not related to the formation of disulfide bond but rather to a difference in physicochemical properties of the stapled peptides.

  1. Figure 1c shows the CD spectrum of peptide 6337 to confirm that the stapled structure promotes the alpha-helix formation. As this becomes the main premise of the paper for interpretation of the rest of the experiments and results, the same CD measurements should be performed and presented for all the other peptides as well (particularly important to compare with the peptide 6579 that has no stapled structure). If the data is too big to be shown in the main figure, the authors might be able to put it in the Supplementary Information.

The reviewer is right. We added the CD spectra for all the peptides. The results confirmed the importance of the stapling effect on the increase of the alpha-helix structure content of the stapled peptides in comparison with the linear one. The figure 1c is now completed with the CD spectra of all the peptides. Discussion of the results is highlighted in yellow in the manuscript (lines 244-250, page 6)

  1. In Figure 4, the cells treated by peptide 6582 and Lipofectamine (those groups that show the highest siRNA efficacies) look much unhealthier than the other groups. Authors need to explain this, probably in relation to the data (cell viability) presented in Figure 5.

We agree with the reviewer. We changed the figure 4 by providing better and higher resolution confocal images where the treated cells look much healthier than in the previous images.

  1. In Figure 5, what was the dosage of siRNA in the lipofectamine group (positive control). It should be clearly mentioned and discussed in the text and the figure caption, because the results from the peptide show dose-dependent behaviors.

The siRNA concentration used with lipofectamine transfection, is 50 nM  and this sentence is now added in the Figure 5 caption. The lipofectamine transfection was only used to test whether the siLuc was effective and we used the protocol conditions provided by the manufacturer. However, it was important to test the stapled peptide/siLuc complexes in a dose-dependent manner to demonstrate their efficacy in gene silencing.

Minor:

  1. Line 38: The first appearance of abbreviation needs the full name description. CPP -> Cell Penetrating Peptide.

The reviewer is right, the full name “Cell Penetrating Peptides” is now added”

  1. Table 3b: comma (,) needs to be replaced by (.) in numbers

As requested by the reviewer, the commas (,) are now replaced the dots (.) in table 3b

Reviewer 2 Report

This manuscript entitled “Hydrocarbon-stapled peptide based-nanoparticles for siRNA delivery” reports the design and synthesis of stapled peptides that promote uptake of siRNA.  The delivery of siRNA to specific tissues and subsequently into the cells of those tissues remains a challenging hurdle for the development of siRNA-based therapeutics.  This manuscript reports a method to overcome the latter.

Minor comments:

  1. In the materials and methods, the sequence of the anti-firefly luciferase is reported. As written, this is not a standard siRNA 19+2 format as there is an extra C on the 5’ end of the sense strand.  If this is a mistake, it should be corrected.  If not, it should be explained.
  2. The different bands (siRNA and siRNA/peptide complex) should be labeled in figure 3a.
  3. The changes in cell morphology in figure 4 require comment and explanation. The cells treated with siRNA complexed with JMV6580 look much healthier than those treated with siRNA complexed with either JMV 6582 or JMV6583.
  4. Discussion as to the lack of correlation of activity presented in figure 5 to uptake shown in figure 2b is missing.

Author Response

We thank the reviewer for his valuable comments and suggestions who helps us to improve our manuscript. Please Find below a point-by-point to your comments and suggestions.

Comments and Suggestions for Authors

This manuscript entitled “Hydrocarbon-stapled peptide based-nanoparticles for siRNA delivery” reports the design and synthesis of stapled peptides that promote uptake of siRNA.  The delivery of siRNA to specific tissues and subsequently into the cells of those tissues remains a challenging hurdle for the development of siRNA-based therapeutics.  This manuscript reports a method to overcome the latter.

Minor comments:

  1. In the materials and methods, the sequence of the anti-firefly luciferase is reported. As written, this is not a standard siRNA 19+2 format as there is an extra C on the 5’ end of the sense strand.  If this is a mistake, it should be corrected.  If not, it should be explained.

Many thanks for this remark, it’s a mistake. A guanine (G) is missing at the 3’ end of the anti-sense strand which is now added. The correction is highlighted in yellow in the text.

  1. The different bands (siRNA and siRNA/peptide complex) should be labeled in figure 3a.

As requested by the reviewer, the different bands were now labelled in figure 3a.

  1. The changes in cell morphology in figure 4 require comment and explanation. The cells treated with siRNA complexed with JMV6580 look much healthier than those treated with siRNA complexed with either JMV 6582 or JMV6583.

We agree with the reviewer and for this we changed the figure 4 by providing better and higher resolution confocal images where the treated cells look much healthier than in the previous images.

  1. Discussion as to the lack of correlation of activity presented in figure 5 to uptake shown in figure 2b is missing.

This point is now added and discussed (Lines 333-337, page 8).

Reviewer 3 Report

The paper by Simon et al. describes the application of hydrocarbon-stapled peptides for siRNA complexation and intracellular delivery. The paper is well written. The concept of using specifically designed peptides for siRNA delivery is interesting and relevant. On the other hand, the manuscript at this point seems premature and would require additional experimentation in support of the claims made. Please find my comments and suggestions below:

Figure 1c: also a control CD spectrum of a non-stapled (linear) peptide should be demonstrated. Preferably show the spectra of all the peptides investigated to compare.

Page 6: it is mentioned that a cysteine was added at the C-terminal end to conjugate the stapled peptides to different cargo molecules. However, the manuscript does not describe any covalent cargo coupling.

Figure 2: JMV6337 is more toxic than the other peptides, which according to the authors is ascribed to the absence of dimerization. First, please explain how dimerization would influence the toxicity profile. Second, please experimentally show (absence of) dimerization of the different peptides with a suitable technique to support this hypothesis. Third, it is known that cytotoxicity of uncomplexed oligo-or polycations is typically higher than the resulting nucleic acid complexes. It would therefore also be of interest to evaluate this peptide for complex formation and siRNA delivery. This could also give the opportunity to evaluate the impact of dimerization on complexation and delivery.

page 7, third paragraph: the authors refer to “gene delivery”, while this paper describes siRNA delivery. Maybe the statement can be generalized to “nucleic acid delivery”

Figure 3: the particles have an extremely high zeta-potential, much higher than typical values that are reported for cationic nanomedicines. Please explain.

Figure 3: the complexes have a high polydispersity. It would be informative to show the size distributions to see where these PDI values arise from.

Figure 3 and others: why didn’t the authors test different N:P values? Varying N:P can have a profound impact on both the physicochemical properties of the complexes (e.g. size), but also the biological activity and toxicity. In my view, it is difficult to conclude on the best performing peptide complex if only one N:P value is tested.

Figure 4: to have a better view on the fluorescent signals and how they differ between the different formulations, I would advise to quantify both FITC fluorescence and Cy5 fluorescence (e.g. by flow cytometry)

Figure 4: the resolution of the fluorescence images is insufficient to make statements on the intracellular distribution of the delivered siRNA. Higher resolution confocal images would be required for this.

Figure 4: the cells treated with JMV6582 appear more rounded/less stretched, comparable to Lipo and in contrast to e.g. JMV6580. This might indicate the initiation of cell death (e.g. apoptosis), while the MTT assay reveals that only Lipo is toxic. Can the authors comment on this?

Figure 5: Luciferase knockdown is compared to non-treated cells, while it should be normalized to luciferase signals obtained with the same formulation, but with siCTRL instead of siLUC. In my view, this is essential to have a reliable quantification of RNAi-mediated knockdown.

Figure 5: the Lipo condition is toxic and does not provide the luciferase knockdown levels that can be expected from Lipofectamine RNAiMAX. Did the authors optimize the positive control?

The paper could benefit from some additional discussion on the intracellular mode-of-action of the peptides (cell internalization, trafficking, endosomal escape).

Author Response

We thank the reviewer for his valuable comments and suggestions who helps us to improve our manuscript. Please Find below a point-by-point to your comments and suggestions.

Comments and Suggestions for Authors

The paper by Simon et al. describes the application of hydrocarbon-stapled peptides for siRNA complexation and intracellular delivery. The paper is well written. The concept of using specifically designed peptides for siRNA delivery is interesting and relevant. On the other hand, the manuscript at this point seems premature and would require additional experimentation in support of the claims made. Please find my comments and suggestions below:

Figure 1c: also a control CD spectrum of a non-stapled (linear) peptide should be demonstrated. Preferably show the spectra of all the peptides investigated to compare.

The reviewer is right. We added the CD spectra for all the peptides. The results confirmed the importance of the stapling effect on the increase of the alpha-helix structure content of the stapled peptides in comparison with the linear one. The figure 1c is now completed with the CD spectra of all the peptides. (also see answer to review 1).

Page 6: it is mentioned that a cysteine was added at the C-terminal end to conjugate the stapled peptides to different cargo molecules. However, the manuscript does not describe any covalent cargo coupling.

Indeed, in this study, no covalent cargo coupling was explored. The Cys was added in order to have a potential point of derivatization on the vectors.

Figure 2: JMV6337 is more toxic than the other peptides, which according to the authors is ascribed to the absence of dimerization. First, please explain how dimerization would influence the toxicity profile. Second, please experimentally show (absence of) dimerization of the different peptides with a suitable technique to support this hypothesis. Third, it is known that cytotoxicity of uncomplexed oligo-or polycations is typically higher than the resulting nucleic acid complexes. It would therefore also be of interest to evaluate this peptide for complex formation and siRNA delivery. This could also give the opportunity to evaluate the impact of dimerization on complexation and delivery.

As requested by the reviewer, to explore whether the absence of cytotoxicity is due to the presence of the cysteine residue and therefore to its dimerization, we completed the profile of cell viability, presented in Figure 2a, in reducing conditions. As shown in Figure S1, no cytotoxicity was observed in reducing conditions (see Figure S1 in Supplementary Material). We concluded from this experiment, that the absence of cytotoxicity is not related to the dimerization but to a difference in physicochemical properties of the stapled peptides. This result is now included in the text (Lines 270-278, page 7).

In addition, the stapled peptide JMV6337 was evaluated for its ability to both encapsulate the siRNA and inhibit the targeted gene when delivered inside the cells. The results showed that JMV6337 was able to encapsulate the siRNA molecule as shown in Figure S2 (Supplementary Material). However, even complexed to siRNA, this peptide is toxic and has no appreciated luciferase inhibitory effect as shown in Figure S3 (Supplementary Material).

page 7, third paragraph: the authors refer to “gene delivery”, while this paper describes siRNA delivery. Maybe the statement can be generalized to “nucleic acid delivery”.

As recommended by the reviewer, “gene delivery” is now replaced by “nucleic acid delivery”

Figure 3: the particles have an extremely high zeta-potential, much higher than typical values that are reported for cationic nanomedicines. Please explain.

This experience was repeated three times and each time we obtained the same high values. Unfortunately, we have no reasonable explanation for this result.

Figure 3: the complexes have a high polydispersity. It would be informative to show the size distributions to see where these PDI values arise from.

We agree that we obtained a high PDI. As shown in the figure below, we can see formation of aggregates in size distribution and in correlogram plots. This figure is only shown for the reviewer and eventually in Supplementary Material if the reviewer considers that it is necessary.

Figure 3 and others: why didn’t the authors test different N:P values? Varying N:P can have a profound impact on both the physicochemical properties of the complexes (e.g. size), but also the biological activity and toxicity. In my view, it is difficult to conclude on the best performing peptide complex if only one N:P value is tested.

We understand the question of the reviewer. Nevertheless, the gel retardation assay presented in figure 3a was performed to determine at which ratio N/P the siRNA is completely encapsulated. Using lower N/P will be useless since we still have free siRNA known to be ineffective alone to enter the cells. Otherwise, using higher N/P will not give better results since we will have excess of stapled peptide. This peptide excess will not be effective for siRNA delivery.

Figure 4: to have a better view on the fluorescent signals and how they differ between the different formulations, I would advise to quantify both FITC fluorescence and Cy5 fluorescence (e.g. by flow cytometry)

As requested by the reviewer, the quantification of the uptake of different formulations are now completed by using flow cytometry. The figure of this quantification is added in the Supplementary Material as Figure S4.

Figure 4: the resolution of the fluorescence images is insufficient to make statements on the intracellular distribution of the delivered siRNA. Higher resolution confocal images would be required for this.

We agree with the reviewer and for this we changed the figure 4 by providing better and higher resolution confocal images.

Figure 4: the cells treated with JMV6582 appear more rounded/less stretched, comparable to Lipo and in contrast to e.g. JMV6580. This might indicate the initiation of cell death (e.g. apoptosis), while the MTT assay reveals that only Lipo is toxic. Can the authors comment on this?

The stapled peptide JMV6582 is not toxic as shown in figure 2a and figure 5. This confocal microscopy experiment was done four times. We provided much better images for the stapled peptides/siRNA formulations in figure 4. The toxicity of lipofectamine is well-known and was reported several times in the literature.

Figure 5: Luciferase knockdown is compared to non-treated cells, while it should be normalized to luciferase signals obtained with the same formulation, but with siCTRL instead of siLUC. In my view, this is essential to have a reliable quantification of RNAi-mediated knockdown.

As requested by the reviewer, the luciferase activity by using siCtrl was added and available in the Supplementary Material as Figure S5. This experiment confirms the specific activity of siLuc in inhibiting luciferase activity induced by the stapled peptides/siRNA formulations.

Figure 5: the Lipo condition is toxic and does not provide the luciferase knockdown levels that can be expected from Lipofectamine RNAiMAX. Did the authors optimize the positive control?

The siLuc lipofectamine tranfection was used as a positive control to test if our designed siLuc is effective to inhibit luciferase activity. For this, we didn’t optimize its experimental condition. We just used the procedure given by the manufacturer.

The paper could benefit from some additional discussion on the intracellular mode-of-action of the peptides (cell internalization, trafficking, endosomal escape).

We agree with the reviewer, the raised points are not explored in our study. However, the confocal microscopy images in Figure 4 display a uniform distribution in the cytoplasm of the green fluorescence related to the stapled peptides which seems to be in agreement with a direct translocation rather than an endocytosis pathway. Obviously, this aspect of intracellular mode-of-action will be investigated in the future.

Round 2

Reviewer 1 Report

The authors diligently addressed the concerns and questions raised by the reviewers. The manuscript has been improved significantly.